# Extracting temporal relationships between weakly coupled peptidergic and motoneuronal signaling: Application to *Drosophila* ecdysis behavior

**Miguel Piñeiro**[1☯], **Wilson Mena**[1,2☯], **John Ewer**[1,3]*, **Patricio Orio**[1,3]*

**1** Centro Interdisciplinario de Neurociencia de Valparaíso, Universidad de Valparaíso, Valparaíso, Chile, **2** Department of Neuroscience, Institut Pasteur, Paris, France, **3** Instituto de Neurociencias, Facultad de Ciencias, Universidad de Valparaíso, Valparaíso, Chile

☯ These authors contributed equally to this work.
* john.ewer@uv.cl (JE); patricio.orio@uv.cl (PO)

**Data Availability Statement:** csv files containing data series analyzed in this work are available at https://github.com/vandal-uv/drosophila_ecdysis.

## Abstract

Neuromodulators, such as neuropeptides, can regulate and reconfigure neural circuits to alter their output, affecting in this way animal physiology and behavior. The interplay between the activity of neuronal circuits, their modulation by neuropeptides, and the resulting behavior, is still poorly understood. Here, we present a quantitative framework to study the relationships between the temporal pattern of activity of peptidergic neurons and of motoneurons during *Drosophila* ecdysis behavior, a highly stereotyped motor sequence that is critical for insect growth. We analyzed, in the time and frequency domains, simultaneous intracellular calcium recordings of peptidergic CCAP (crustacean cardioactive peptide) neurons and motoneurons obtained from isolated central nervous systems throughout fictive ecdysis behavior induced *ex vivo* by Ecdysis triggering hormone. We found that the activity of both neuronal populations is tightly coupled in a cross-frequency manner, suggesting that CCAP neurons modulate the frequency of motoneuron firing. To explore this idea further, we used a probabilistic logistic model to show that calcium dynamics in CCAP neurons can predict the oscillation of motoneurons, both in a simple model and in a conductance-based model capable of simulating many features of the observed neural dynamics. Finally, we developed an algorithm to quantify the motor behavior observed in videos of pupal ecdysis, and compared their features to the patterns of neuronal calcium activity recorded *ex vivo*. We found that the motor activity of the intact animal is more regular than the motoneuronal activity recorded from *ex vivo* preparations during fictive ecdysis behavior; the analysis of the patterns of movement also allowed us to identify a new post-ecdysis phase.

## Author summary

Repetitive movements such as walking, swimming, and flying are controlled by networks of neurons known as central pattern generators. In many cases, the exact pattern of

**Funding:** This work was supported by Fondecyt Grants 1181076 (to PO) and 1180403 (to JE) and the Advanced Center for Electrical and Electronic Engineering - ANID (FB0008 to PO). Powered@NLHPC: This research was partially supported by the supercomputing infrastructure of the NLHPC (ECM-02). The Centro Interdisciplinario de Neurociencia de Valparaíso (CINV) is a Millenium Institute supported by the ANID grant ICN09_022. The funders had no role in study design, data collection and analysis, decision to publish, or preparation of the manuscript.

**Competing interests:** The authors have declared that no competing interests exist.

activity is modulated by neuropeptides, which are small signaling molecules that, unlike neurotransmitters, are broadly released within regions of the nervous system. Because of their mode of action, it can be difficult to discern the relationship between the temporal pattern of firing of peptidergic neurons and the timing of the resulting motor behavior. Here, we developed methods to analyze the patterns of activity of such weakly coupled systems as applied to ecdysis, the stereotyped sequence of behaviors used by insects to shed the remains of their old exoskeleton at the end of every molt. Key actors in this process are motoneurons (MN) and a set of neurons expressing the neuropeptide, Crustacean Cardioactive Peptide (CCAP). Combining calcium imaging, frequency analysis, computational simulations, and image processing, we determined the relationships between the activity of CCAP neurons and the resulting motor output during pupal ecdysis in the fruit fly, *Drosophila melanogaster*. We found that several temporal features of the activity of CCAP neurons are highly coupled to the pattern of motoneuronal activity, suggesting an active role of CCAP neurons during ecdysis. We also developed quantitative approaches that allowed us to identify a new sub-phase of ecdysis behavior.

## Introduction

Oscillatory neural circuits are important for many brain processes including memory formation [1,2], sensory representation [3–6], and rhythmic pattern generation [7,8]. Rhythmic movements are controlled by neuronal networks that time the firing of motoneuron discharges, which then cause a sequence of organized movements. In order to generate organized behaviors, it is necessary to dynamically coordinate the interaction of local and sparse brain circuits [9,10]. How these dynamic properties are tuned can profoundly influence the functional connectivity that defines the structure of neural circuits that orchestrate behaviors. In this context, neuromodulators such as neuropeptides have been shown to play a major role in regulating and coordinating network functions in a number of processes including feeding, sleep, courtship, stress, learning and memory, amongst others [11–16].

Centrally coordinated innate behaviors have provided a useful model to study the molecules, neurons, and networks that organize sequential and rhythmic behaviors. One innate behavior that has been used for these studies is insect ecdysis, which is a stereotyped sequence of three motor programs (pre-ecdysis, ecdysis itself, and post-ecdysis) that is required to shed the remains of the old cuticle (exoskeleton) at the end of each molt [17–19]. Multiple neuropeptidergic circuits have been implicated in the regulation of the ecdysis but their precise roles are still poorly understood. Ecdysis begins with the release of the neuropeptide Ecdysis Triggering Hormone (ETH), into the circulatory system (hemolymph). ETH is synthesized and released from peripheral endocrine Inka cells [20]. Once ETH reaches the Central Nervous System (CNS) it sequentially activates several neuropeptidergic targets, where the network expressing the Crustacean Cardioactive Peptide (CCAP) has been suggested to be a critical node for the generation of the ecdysial motor pattern [21–27]. However, the mechanisms by which the pattern of activity of the CCAP network is then translated into a motor output are not fully understood.

Recent advances in imaging technology enable the recording of neuronal activity in large regions of the brain including hundreds of neurons, thereby providing new ways to study circuit dynamics and behavior. Nevertheless, it remains challenging to extract quantitative information from such large data sets. It is thus necessary to develop suitable algorithms to determine the time windows during which specific motor activity occurs, and to identify the

neurons that show activity associated with the initiation and termination of a motor pattern. Previous approaches have used different methods to quantitatively classify neuronal activity patterns, which include principal components analysis (PCA), independent components analysis (ICA), singular-value decomposition (SVD), and k-means clustering, [28–30]. These methods have been widely used but their application is often subject to dataset-specific parameters and interpretations. Therefore, the generation of methods that can be more widely applied would be of great utility to the field.

Here, we report on the implementation of computational approaches to decode the signal dynamics driving ecdysis in the fruitfly, *Drosophila melanogaster*. We used mathematical methods and models to simultaneously analyze calcium imaging of CCAP neurons and motoneuron activity during the behavior. Although the pattern and the timing of activity of these two populations of neurons differed significantly, we were able to show that the activity of CCAP neurons is functionally tightly coupled to that of the motoneurons during the ecdysis and post-ecdysis phases, in a cross-frequency manner. This was further proved by a probabilistic logistic model fitted to the experimental data, that from the CCAP neuron activity can predict the times at which motoneurons had a high chance of oscillating. We also generated a conductance-based model that simulates many of the observed experimental features. Finally, we developed an algorithm that extracts the major traits of ecdysis behaviors, allowing us to quantify the movements that occur during the behavior of the intact animal and contrast them with the *ex vivo* recorded motoneural activity. This algorithm also allowed us to identify a new sub-phase within the post-ecdysis period. In summary, we describe a series of methods to quantify and correlate patterns of neuronal activity with differing temporal characteristics that occur during the expression of a stereotyped behavior. Using these methods, we show that the CCAP network tightly regulates motoneuronal activity throughout the execution of the entire ecdysis and post-ecdysis routines.

## Results

### Individual dynamics of CCAP-expressing neurons and motoneurons

The pattern of neural activity that corresponds to ecdysis behavior can be elicited in *ex vivo* preparations of *D. melanogaster* CNS by exposure to ETH. We followed this approach using CNSs from animals just prior to pupal ecdysis, which expressed the genetically-encoded calcium sensor GCaMP3.2 (as a proxy for neural activity), either in CCAP neurons or in both CCAP neurons and motoneurons (Fig 1). As has previously been reported [22,25], increases in GCaMP signal typically began 20 minutes after stimulation with ETH, around the time that the ecdysis phase is induced in intact pupae. During this phase, CCAP neurons and motoneurons display higher levels of activity, which then falls after entering the post-ecdysis phase.

CCAP neurons can be divided into α and β neurons depending on their location and their activity pattern [25]. Here, we mostly focused on the α type, because β neurons appeared to display a low pass-filtered version of the activity of the α type. Regarding motoneurons, we detected that the activity on each side of the animal is highly correlated (S1 Fig). Thus, we divided them into left and right regions and calculated the average motoneuronal activity for each side. This procedure also increased the signal-to-noise ratio (SNR), making the task of comparing different experiments easier.

In the intact animal, the onset of left-right alternating motoneuronal activity corresponds to the beginning of the ecdysial phase of the ecdysis motor sequence, after pre-ecdysis has finished [22,25]. As a first approach to characterize the dynamics of CCAP neuron and motoneuron activity, we computed the onset of their activity in preparations in which both classes of neurons expressed the GCaMP3.2 calcium sensor (see *Methods*, section *Activity onset*). The

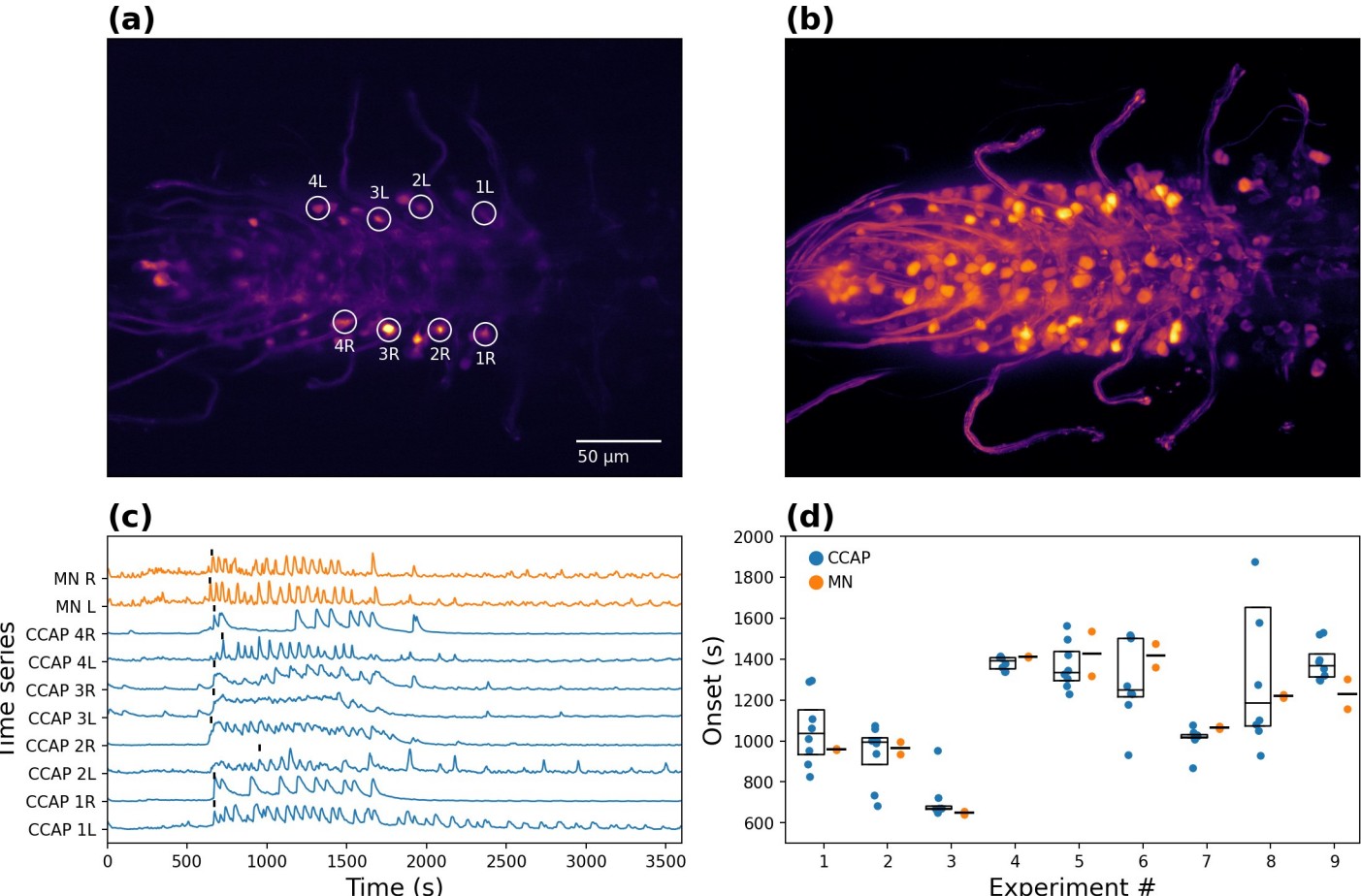

**Fig 1. CCAP neuron and motoneuron activity. (a)** Single-plane calcium imaging of GCaMP3.2-expressing CCAP neurons. **(b)** Projection of 5 images from different planes, of GCaMP3.2-expressing CCAP neurons and motoneurons. The same scale bar from panel (a) applies. **(c)** Time series of signals from calcium sensor in AN1-AN4 α CCAP neurons and motoneurons recorded in a single experiment; time zero corresponds to moment of ETH-stimulation. The letter indicates the side (left [L] and right [R]), while the number indicates the abdominal segment of the neurons. Vertical marks denote the onset of oscillatory activity, as detected by the procedure indicated in Methods. **(d)** Mean time of onset of α CCAP neuron and motoneuron activity, for each of 9 separate experiments, showing temporally close values between populations. "MN" and "CCAP" indicate motoneurons and α CCAP neurons, respectively. Each MN-CCAP pair corresponds to a single experiment, with its 2 MN and 8 CCAP recordings. Box plots indicate intra-experiment median and quartiles.

mean onset time after ETH challenge was 1176 ± 37.9 s for the α CCAP neurons and 1149 ± 61.5 s for the motoneurons (Fig 1D) (n = 9, mean ± SE).

The mean onset of activity in α CCAP neurons and motoneurons tended to be temporally correlated. Indeed, for all experiments except one, the onset of activity in α CCAP neurons and in motoneurons was significantly close in time (p-values < 0.01; one-tailed Mann-Whitney U test). In contrast, when comparing the onset time of α CCAP and motoneurons from different experiments, we found that they were more temporally separated. In addition, the onset of motoneuron activity usually occurs after at least two or three CCAP neurons show increases in activity, suggesting that some level of α CCAP activity is required to initiate the motoneuronal oscillatory activity.

Next, we computed the period of the oscillations of both populations of neurons using the continuous wavelet transform (CWT). This method was preferred over the Fourier transform, as CWT can localize the frequency components in time. The average scaleograms for all α CCAP neurons (n = 111) (Fig 2A) and motoneuron time series (n = 18) (Fig 2B) showed that

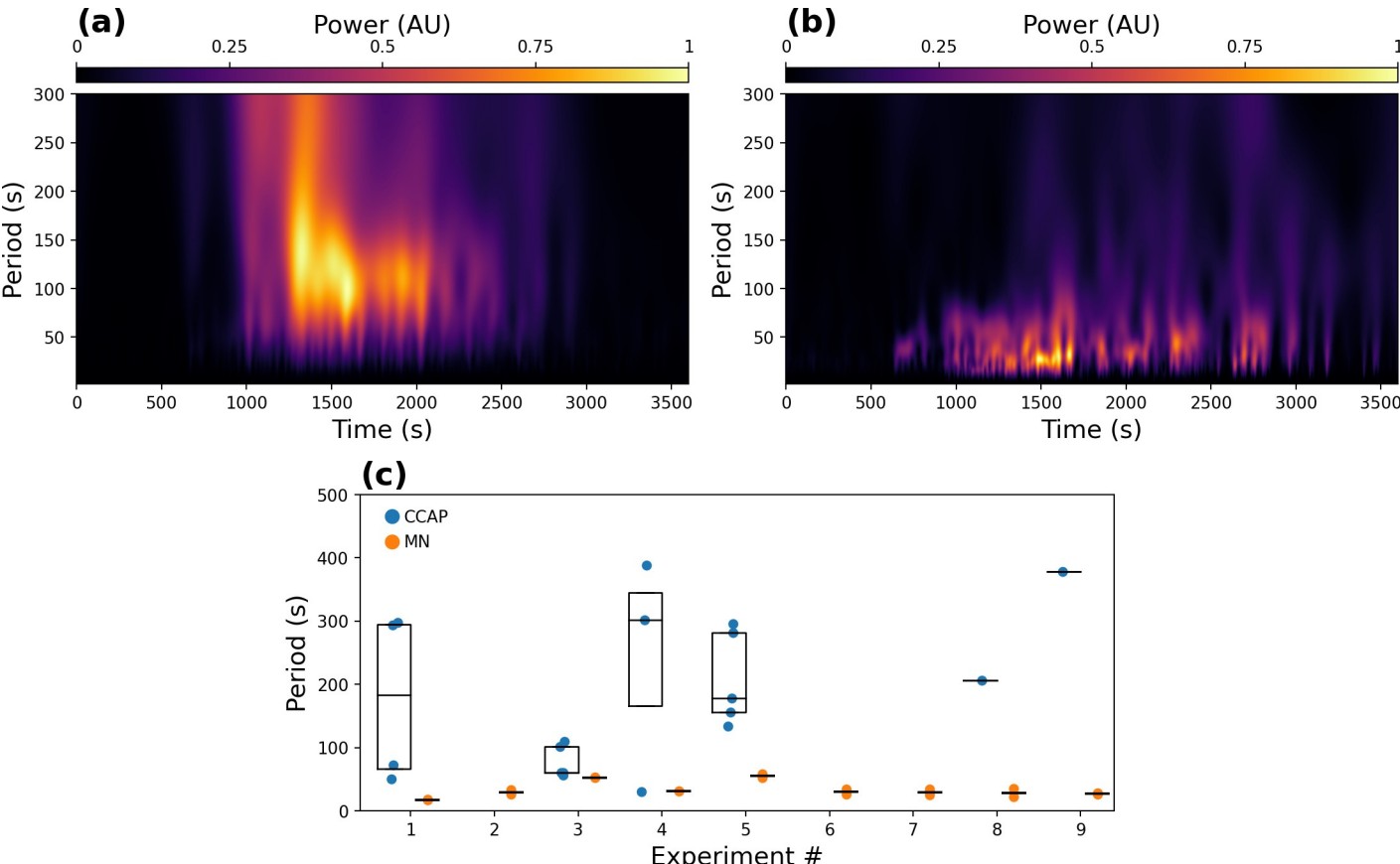

**Fig 2. Oscillation period of CCAP neurons and motoneurons. (a, b)** Average scaleogram for all α CCAP neuron (**a**) and motoneuron time series (**b**). **(c)** Mean oscillation periods of CCAP neurons and motoneurons for all 9 experiments. For CCAP neurons, only the data for neurons that passed the selection criteria are shown (see text). In some experiments none were accepted, and the data are missing. By contrast, in all experiments motoneurons of the left and right regions showed a dominant oscillation period.

the main oscillatory period was around 50 to 200 s for α CCAP neurons, and around 25 to 50 s for motoneurons.

Nevertheless, not all α CCAP neurons showed a clear single dominant frequency component. Applying a criterion for the existence of predominant peaks in the frequency spectrum (that the minimum amplitude at each side of the interval that goes from half to twice the period of the maximum amplitude be less than 80% of the maximum) we found that only 47% of the α CCAP neurons passed the selection criterion, illustrating the irregularity of the oscillatory activity in these neurons. Neurons that passed this criterion were used to compute the mean period for each experiment ($166 \pm 23.3$ s; $n = 13$ experiments); the remaining neurons were eliminated from further frequency analysis.

In contrast to CCAP neurons, motoneurons displayed a more regular pattern of activity with a clearer main oscillatory component, and no motoneuron time-series was discarded. The main oscillatory period of motoneurons was $33.4 \pm 4.1$ s ($n = 9$). The difference in the principal oscillatory period between the left and right sides was also small, suggesting that the activity of motoneurons on both sides is not independent. We plotted the periods of both CCAP and motoneurons and their means (Fig 2C), and in all experiments found that the period of motoneuron activity was much shorter than that of CCAP neurons.

## Coordination between CCAP neurons

To investigate the coordination between CCAP neurons, we measured the linear relationship between their time series using Pearson's correlation. We grouped the correlation pairs into functionally equivalent pairs based on what is known about their anatomy, as well as their synaptic and peptidergic connectivity [31,32]. Thus, the correlation pairs were grouped into: contralateral neurons (on the same segment but opposite sides), ipsilateral neurons (on the same side but in different segments), and "other" neuron pairs (on opposite sides and on a different segments) (see Fig 3A). We computed the group p-values using the one-tailed Mann-Whitney U test to compare correlations for each experiment with null cross-experiment correlations. In all cases we obtained p-values < 0.001 making group correlations shown in Fig 3B–3F highly significant.

Correlations values for contralateral CCAP neurons were higher than for ipsilateral neurons, for αα and αβ pairs (Fig 3B–3D). The correlations values for contralateral neurons were also higher than for "other" neurons for αα, ββ and αβ pairs. To test if the distance (in

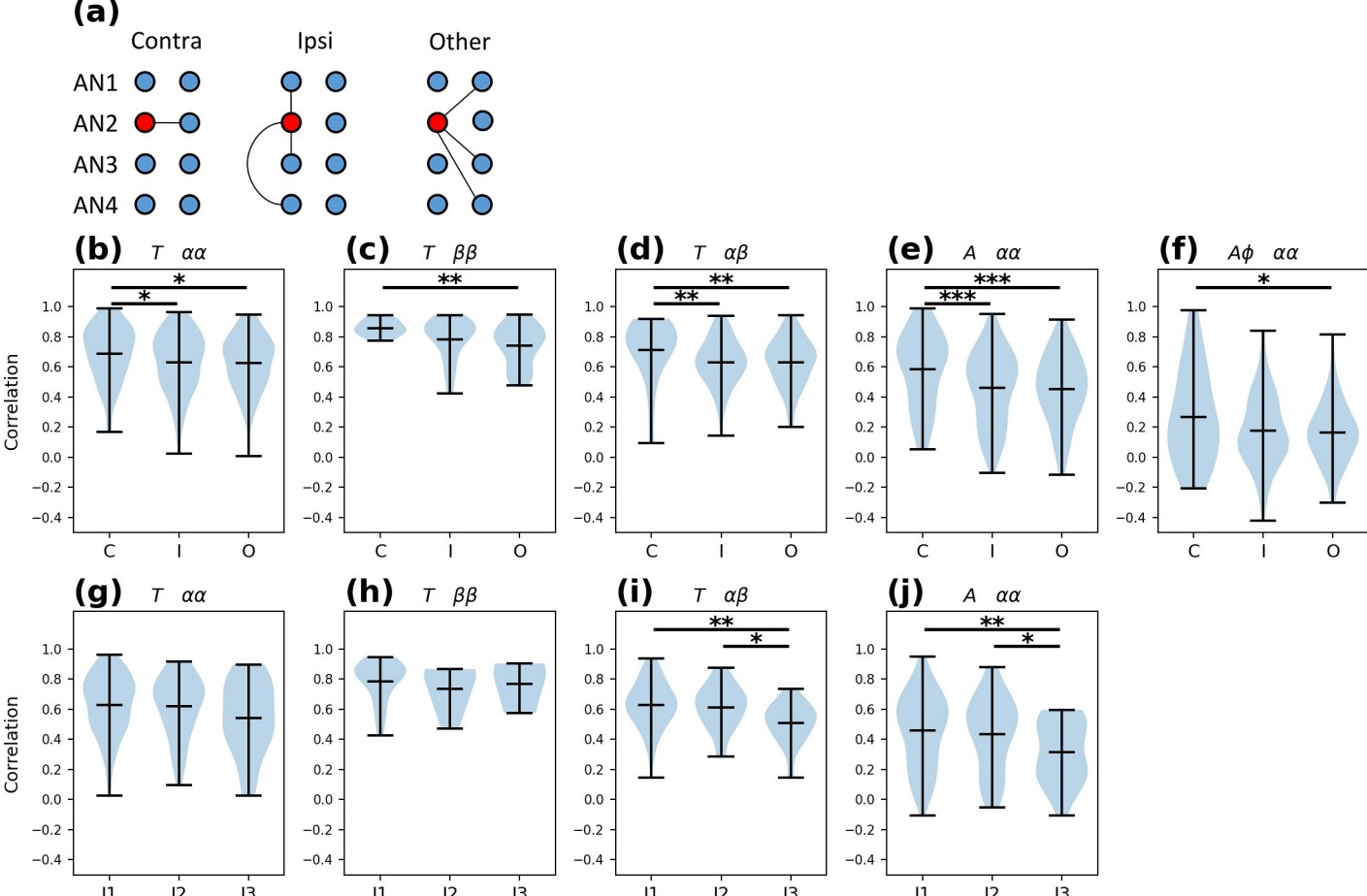

**Fig 3. CCAP neuron coordination.** (a) Pearson's correlations pairs are divided into 3 groups: contralateral neurons ("C"), ipsilateral neurons ("I"), and "others" ("O"). As an example, the graph shows pairs that include the left AN2 neuron. (b-j) Correlation coefficients between the time series of CCAP neurons, shown as violin plots with their minimum, maximum and mean values. (b) α CCAP neuron correlations in the time domain (T). (c) β CCAP neuron correlations in the time domain. (d) αβ CCAP neuron correlations in the time domain. (e) Correlations of α CCAP neuron amplitude of oscillations (A) in the time-frequency domain. (f) Correlation of α CCAP neuron amplitude and phase (Aϕ) in the time-frequency domain. (g-j) Correlation between ipsilateral pairs with different segmental separation, "I1", "I2" and "I3" groups are contiguous, separated by 1 and separated by 2 segments, respectively. The plots use the same notation as the plots in (b-f). Data were compared using a non-parametric Mann-Whitney U test. *: p-value < 0.05, **: p-value < 0.01, ***: p-value < 0.001.

segments) between neurons affected the strength of the coupling, we divided the pairs into 3 ipsilateral groups based on their segment separation. The I1, I2 and I3 groups contain ipsilateral pairs of neurons within the same segment, or pairs separated by 1 or by 2 segments, respectively. We performed the same analyses on these groups (Fig 3G–3J), and found that correlation for both αα and αβ neuron pairs dropped as the segmental distance between them increased (Fig 3I–3J). In contrast, coordination between ββ neuron pairs was not affected by segmental distance (Fig 3H).

Finally, we studied the correlation of the αα pairs in the time-frequency domain, using the 50 to 200 s period band of the CWT. This also allowed us to study the correlation of the oscillation's amplitude with and without the phase component. We found that the signal amplitude (absolute value of the CWT) in contralateral neurons showed a higher correlation compared to that of other pairs (Fig 3E). When taking into account the phase component (the real part of the CWT) the correlation coefficients displayed an important drop, showing that the phase of the oscillations was poorly coordinated (Fig 3F). The interpretation of these results is that CCAP neurons tend to be active at the same time but do not oscillate with the same phase.

These results show that all abdominal segments 1–4 (AN1-AN4) CCAP neurons tend to have synchronized activity, but that their coupling strength varies depending on the neuronal pair considered. Contralateral αα and αβ pairs from the same segment appear to show higher coupling, whereas ββ pairs show similar contralateral and ipsilateral coupling strength. Finally, the correlated activity does not involve a synchronized oscillation, i.e. CCAP neurons are active at the same time but not in a coordinated fashion.

## Coordination between motoneurons

Left and right motoneuronal regions express coordinated but opposite activity. In order to quantify this coordination and determine how it evolves over time, we calculated the Pearson's correlation between left and right motoneuron regions on a sliding window of 100 s (thus the window is longer than the oscillation period, but shorter than the duration of oscillation bursts). The correlations tended to be negative during the oscillating periods and positive during the non-oscillating periods (Fig 4A), consistent with the observed synchronous but phase-opposite behavior.

We then used the CWT to compute the instantaneous amplitude and phase of the motoneurons' primary oscillatory period for the entire recording. We generated vectors with angles equal to the phase difference and lengths equal to the mean amplitude of their CWT, computing the phase difference of the regions by averaging all the vectors for a given experiment (Fig 4B).

Using this procedure, we found a large amount of variability in the phase difference within each experiment, and an important spread of the experimental mean phase difference (Fig 4C), with a global mean of 182.1 ± 14.6˚.

The fact that the motoneurons oscillate in antiphase is compatible with the existence of a central pattern generator (CPG) downstream of the CCAP neurons [27]. The variability in the phase difference could be an indicator that the CPG imaged in the calcium imaging preparations has difficulty synchronizing left and right motoneurons in the absence of sensory feedback, analogous to what has been shown previously for *Drosophila* larval crawling behavior [33].

## Functional connectivity between CCAP neurons and motoneurons

We noticed that α CCAP neurons appear to modulate the amplitude of the motoneuronal activity, with high levels of α CCAP neuron fluorescence tending to match periods of

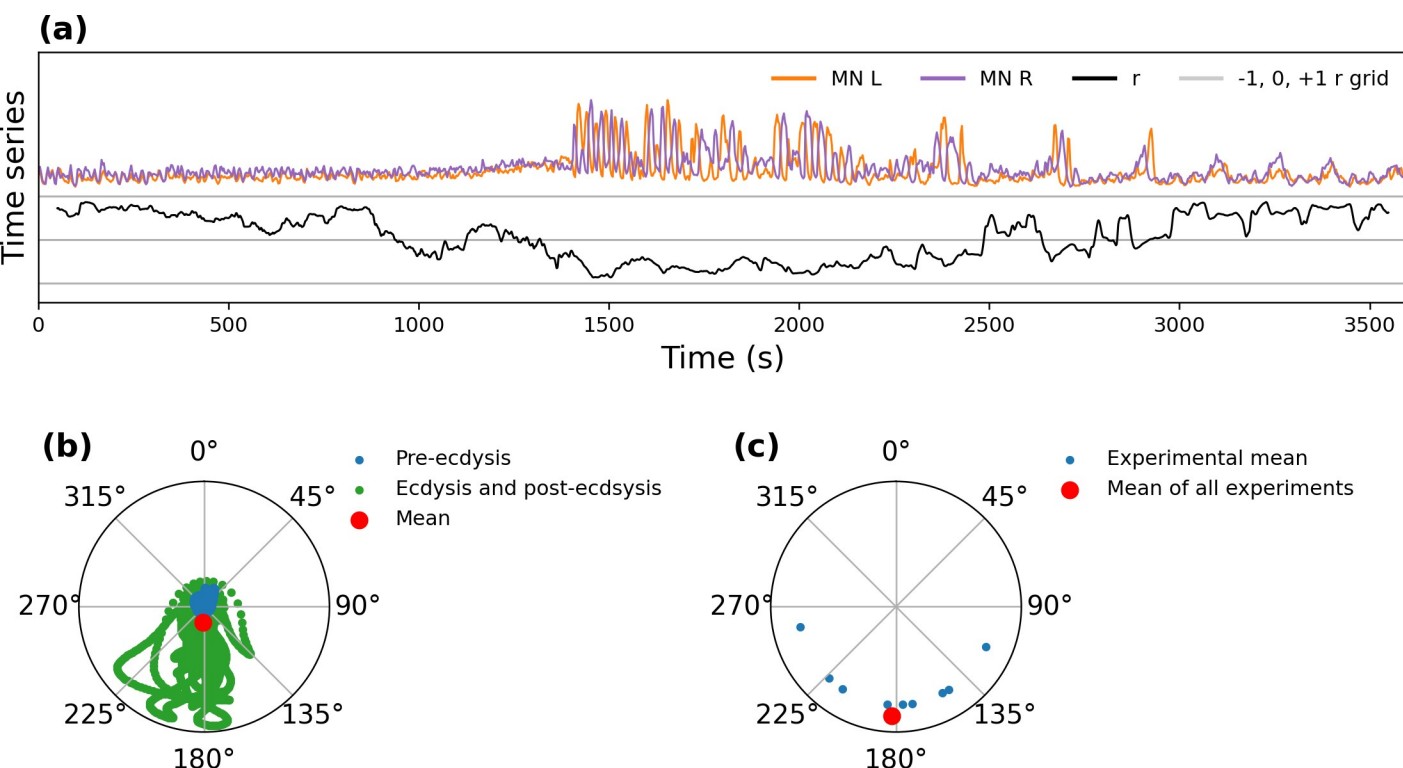

**Fig 4. Coordination between motoneurons. (a)** Correlation coefficients between left and right motoneuron time series, calculated over a sliding window of 100 s. Time series for left ("MN L") and right ("MN R") motoneuronal regions are shown in orange and purple, respectively, and the Pearson's correlation coefficient ("r") of the sliding window is shown in black. Gray horizontal lines indicate correlations of −1, 0 and +1. **(b)** Example of the method used to compute the mean phase difference of an experimental recording. Blue and green points represent the phase difference at every instant of the experiment; their amplitude is scaled to the mean amplitude of the oscillations of the left and right region. The red point represents the mean vector, whose phase represents the mean phase difference of the experiment. **(c)** Mean phase difference for 9 experiments (blue) and mean for all experiments (red).

motoneuronal oscillation. Based on this observation, we converted the motoneuronal signal so that it could be correlated quantitatively to that of CCAP neurons. For this we used a single motoneuronal signal calculated as the difference between the right and left time series. This procedure reduced the common noise and increased the SNR and oscillation amplitude, without much loss of information as the two-time series are mostly redundant.

We computed the absolute value of the CWT of the motoneuronal signal at its previously computed primary oscillatory period, thus extracting the instantaneous amplitude (Fig 5A). The amplitude signal from the motoneuronal oscillation was then correlated to each α CCAP neuron time series using Pearson's correlation. The significance of the within-experiment correlations, as compared to null cross-experiment correlations, was tested using a one-tailed Mann-Whitney U test. For all experiments except one, the correlations were significant (Fig 5B). The one experiment that did not display significant correlations showed post-ecdysis motoneuronal oscillations that did not match temporally the increases in α CCAP neuron activity.

As the time of onset of activity varied across experiments, it could be argued that the high significance of the correlation between α CCAP neurons and motoneurons is caused by the matching of onset times. To test this hypothesis, we removed from each recording the initial non-oscillatory section and repeated the analysis using the modified time series (Fig 5C). For all experiments except two, the results were significant, suggesting that α CCAP neurons regulate the motoneuronal activity during the entire recording period.

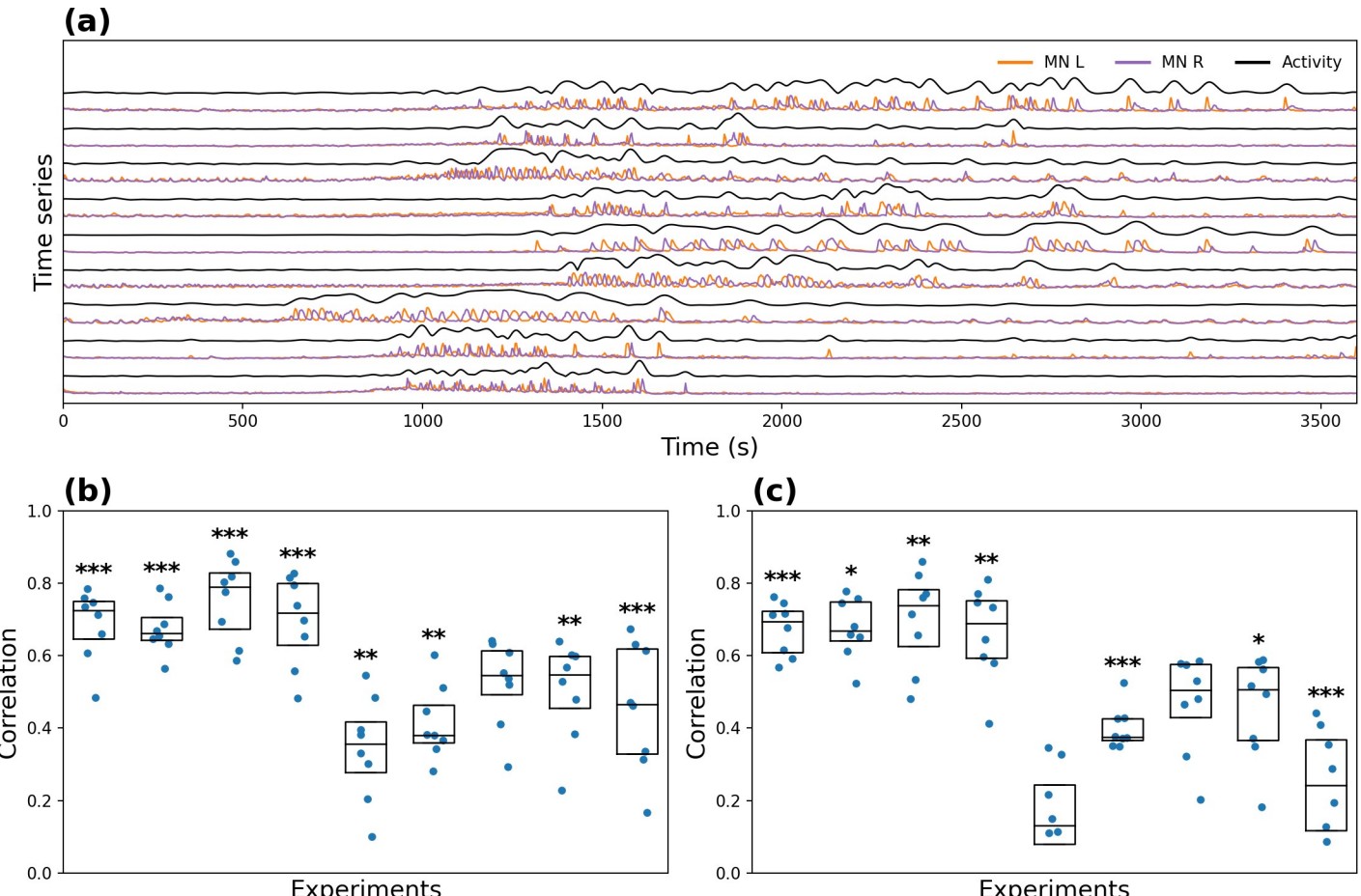

**Fig 5. Correlation between α CCAP neuron and motoneuron activity. (a)** Orange and purple lines show the activity of the left ("MN L") and right ("MN R") motoneuron regions, respectively, and the black line (labeled "Activity") shows the amplitude of the motoneuron signal. **(b)** Correlation coefficients between α CCAP neurons and the amplitude of the motoneuronal time series, with individual data as points and boxes indicating median and quartiles. **(c)** Analogous to **(b)**, but with the pre-ecdysis phase removed. Significance of the within-experiment correlations was tested using a one-tailed Mann-Whitney U test comparing to null cross-experiment correlations *: p-value < 0.05, **: p-value < 0.01, ***: p-value < 0.001.

These results suggest that the CPG responsible for ecdysis requires a constant input from the α CCAP neurons to maintain its ongoing oscillatory activity, consistent with previous findings [21,27]. Also, the correlated activity allows us to talk of *functional connectivity* [34] that occurs between CCAP and motoneurons, regardless of whether there is a structural connectivity (synapses) between them.

## Fitting the CCAP-motoneuron interaction with a logistic model

After finding that CCAP neurons are functionally coupled to the motoneurons, we built a model to test how well the activity of the α CCAP neurons could predict the motoneuronal oscillatory state. The aim of this effort is to advance a step forward from simply observing correlations, as it will test with a quantitative criterion the temporal match between the activity of the different neuronal populations.

The model takes the activity of α CCAP neurons as input and generates the motoneuronal oscillatory activity as output. As the amplitude of the motoneuron oscillations does not appear to be regulated by α CCAP neurons, we employed a binary signal obtained by thresholding the

oscillation amplitude (see Methods) to describe oscillatory and non-oscillatory motoneuronal activity. The motoneuron oscillation generator (CPG) integrates the signals from the AN1-AN4 α CCAP neurons and produces a probabilistic oscillatory response.

The system was modeled as a logistic regression:

$$p(t) = \frac{1}{1 + \exp(-\beta - \sum_{i=1}^{8} w_i f_i(t))}$$

Where $p(t)$ represents the probability that motoneurons will oscillate at time t; $f_i(t)$ the $i$-th α CCAP neuron time series; $w_i$, the weight of the $i$-th α CCAP neuron; and β the offset. The model's coefficients were estimated using maximum likelihood estimation (MLE) with the constraint that all weights must be positive.

As shown in Fig 6A and 6B, the maximum likelihood solution has weights set to zero, i.e., not all the CCAP neuron time series of every experiment are needed to predict the motoneuronal oscillations. The minimum number of α CCAP neurons required to reach the maximum likelihood was 2, the maximum was 6 and the average was around 4. These results should not be interpreted to mean that some α CCAP neurons do not have any effect on the motoneuronal oscillatory activity; rather, we expect the redundancy of their activity to make the likelihood of the model to be maximized with only some of them. The weights were highly variable, also an indicator of the degeneracy in the system, as α CCAP neurons with diverse activity dynamics nonetheless generate similar motoneuronal activity.

Fig 6A shows that during the oscillatory activity of motoneurons, $p(t)$ increased accordingly; one exception can be observed in Fig 6B, where $p(t)$ barely increased during rare post-ecdysis oscillatory events, due to the lack of significant increases in the activity of the α CCAP neurons during this period (this is the same experiment that showed poor correlation between CCAP and motoneuron activity).

We also fitted a single weight model (same value for all $w_i$), based on the assumption that all α CCAP neurons affected motoneuronal activity in the same way. To select the best model by taking into account the tradeoff between the goodness of fit and the complexity of the model we used the Akaike Information Criterion (AIC) [35]. For all 9 experiments, the multi-weight model performed noticeably better (i.e., its values were lower) than did the single-weight model (Table 1). This suggests that not all CCAP neurons have the same impact on the motoneuronal time series; the origin of this could be experimental, biological, or a combination of the two.

## Reproducing motoneuron calcium activity using a conductance-based model

The logistic model is an abstract model that can fit a probability function to a binary motoneuron oscillation time series; however, it is not capable of modeling calcium dynamics. To test if we could reproduce the observed calcium dynamics, we used a model developed by Jalil et al. [36], consisting of two endogenously bursting neurons with fast non-delayed inhibitory connections that synchronize in antiphase. The activity of the neurons depends on the voltage-dependent potassium and sodium currents and on reciprocally inhibitory synapses between them. To couple the oscillatory activity of the model to the CCAP neurons, we added a depolarizing current that depends on the activity of the α CCAP neurons. The model was further adapted to generate fluorescence spikes during the phase of motoneuronal oscillation, matching the oscillation timing, the calcium interspike interval (ISI), the spike phase difference, and the time constant of each experiment (see methods).

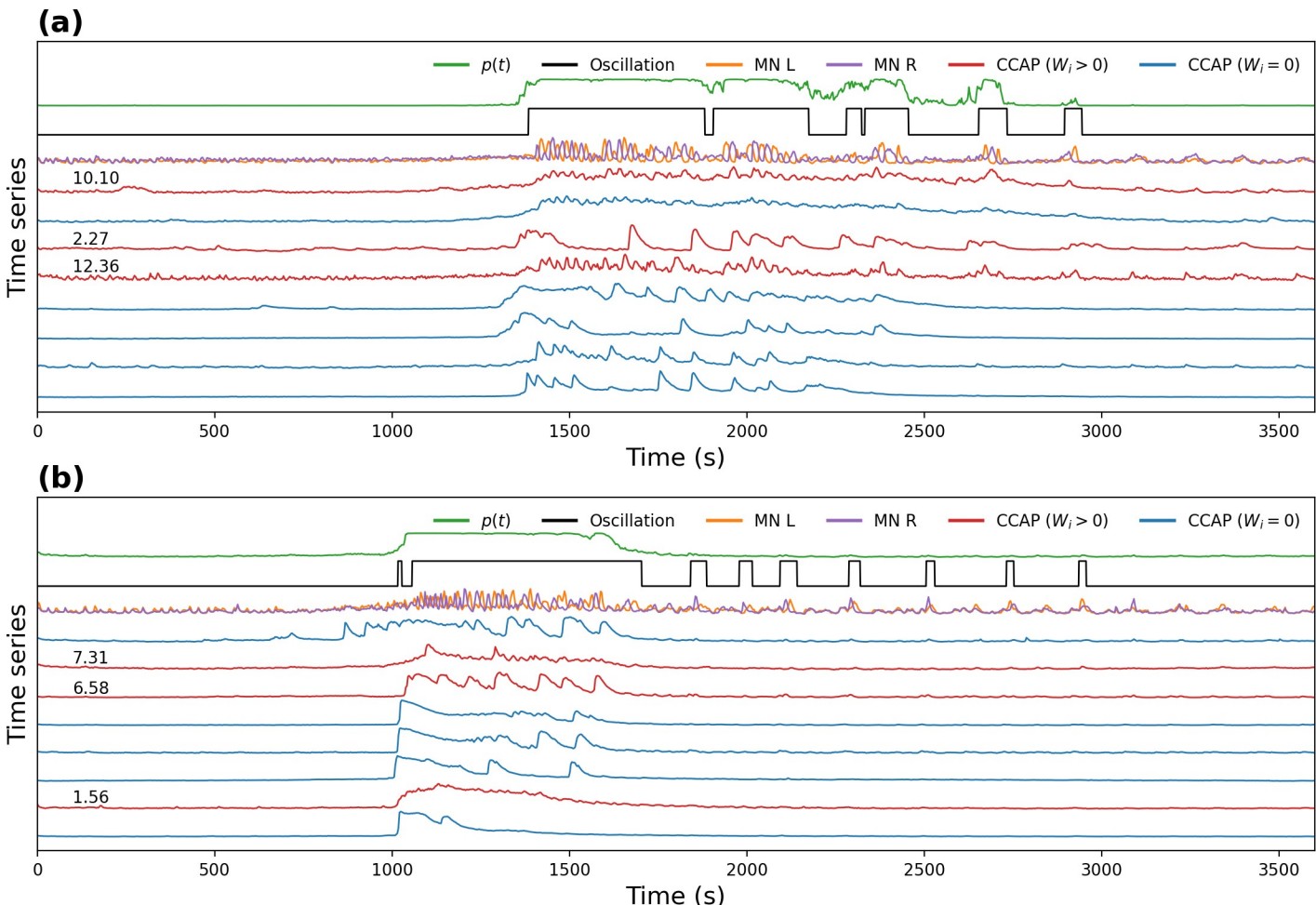

**Fig 6. Fit of the logistic model to the experimental data. . (a, b)** Time series of α CCAP neurons (red, blue), motoneurons (orange, purple), binarized oscillatory activity of motoneurons (black), and probability of oscillation predicted by the model (green). CCAP activity traces are shown in red with their corresponding weight value if it is positive ("CCAP ($W_i > 0$)"), or in blue with no value if it is zero ("CCAP ($W_i = 0$)"). All weight values were computed through multi-weight model fitting. **(a)** Example of a good match between the model $p(t)$ and the oscillatory state of the motoneurons. **(b)** Example of a poor match during the post-ecdysis phase, as a result of the lack of α CCAP activity.

In our model, 8 α CCAP neurons are coupled to 2 motoneurons (Fig 7A) and each α CCAP neuron releases a peptidergic signal that depolarizes the motoneurons and causes them to oscillate. α CCAP neurons release the signal according to the recorded fluorescence time series factored by a weight (already computed in the logistic model). Outputs from the peptidergic neurons are then multiplied by weights, summed, and transformed with a logistic function to generate the gating variable $p(t)$. In this way, α CCAP neurons modulate the oscillatory behavior of the motoneurons through $p(t)$.

**Table 1. Akaike information criterion.** AIC values of single- and multi-weight model fit for every experiment.

| | Experiments | | | | | | | | |
|---|---|---|---|---|---|---|---|---|---|
| | 1 | 2 | 3 | 4 | 5 | 6 | 7 | 8 | 9 |
| Single-weight | 908 | 1803 | 1092 | 1519 | 3714 | 2976 | 2312 | 1489 | 3108 |
| Multi-weight | 711 | 1598 | 888 | 891 | 2481 | 2464 | 2019 | 1227 | 2675 |

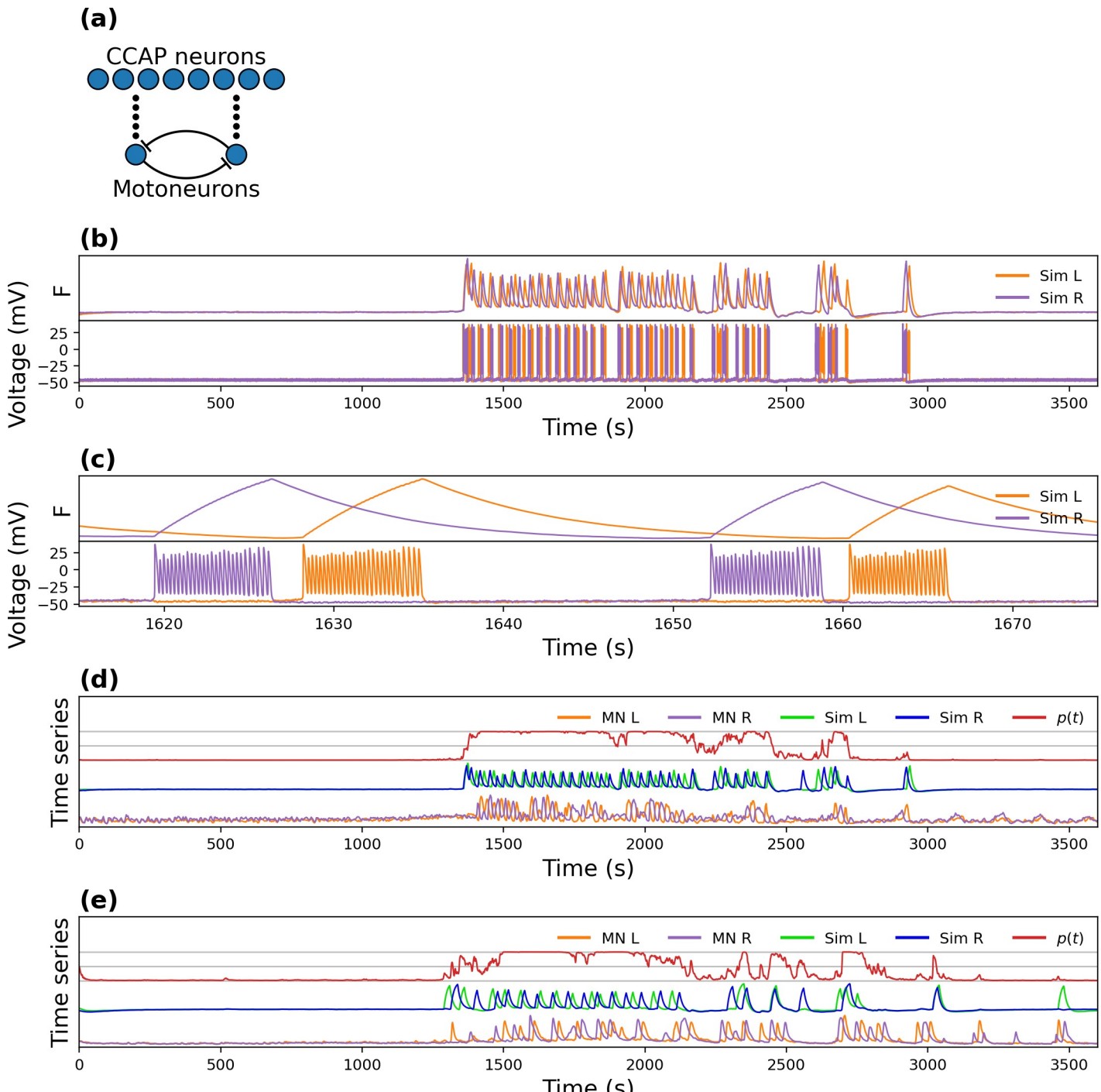

**Fig 7. Simulation of fluorescence spikes. (a)** Model circuit structure, showing the 8 α CCAP neurons that release the peptidergic signal (black circles) and activate the antiphase oscillatory behavior of the left and right motoneurons. **(b)** Simulated fluorescence (top) and voltage (bottom) time series of left ("Sim L") and right ("Sim R") motoneurons. **(c)** Magnification of a small segment of (b). **(d, e)** Simulation of two different experiments. The gray grid marks probabilities of 0, 0.5 and 1. "MN L", "MN R", "Sim L" and "Sim R" indicate experimental left, experimental right, simulated left, and simulated right motoneurons, respectively. $p(t)$ indicates the probability of oscillation.

We fitted the model during multiple passes of manual parameter adjustments and simulation sessions. The resulting parameter values were identical for all experiments except for $\tau_f$ and $\tau_K$, which were used to fit the exponential decay and oscillation period, respectively.

An example of the bursting behavior produced by the model is shown in Fig 7B, where motoneuron fluorescence and voltage are plotted next to each other for comparison. It shows that the simulated fluorescence increases during the bursting phase and decreases during the non-bursting phase. When the model is run using experimental CCAP time series as its inputs, it can reproduce fairly well the motoneuron oscillatory behavior (Fig 7D and 7E). The timing of the simulated oscillations approximately matches that of the experiments, even during the periods of lower spiking frequency. This is especially interesting considering that $p(t)$ was fitted to a binarized (as opposed to a graded) motoneuronal activity signal. Another noteworthy result is that, as more time passes after an oscillation, the neurons are more likely to begin oscillating again. This effect is the result of the slow dynamics of the potassium current, which gradually stops inhibiting action potentials (APs); this can be seen in the last oscillation period in Fig 7D.

The simulation spike frequency and its exponential decay matched that of the experiments, as they were fitted through the $\tau_{Na}$ and $\tau_f$ parameters. The noise caused by Eq (7) adds amplitude and phase variability, resembling the one observed in the experiments.

Fig 8 compares simulated and experimental motoneuron time series for 3 different experiments in the time and time-frequency domains. To reduce the noise of the model scaleogram, each simulation was repeated 10 times and their scaleograms averaged. In all experiments the simulations showed a good time-frequency match to the experimental data.

Even though very little is known about the structure of the circuit, our model replicated many of the features of the activity pattern observed in the experimental recordings, showing that the activity of CCAP neurons is tightly linked to that of motoneurons.

## Correspondence between calcium activity and motor behavior

Finally, we wanted to quantify the extent to which the neural activity recorded in *ex vivo* CNS preparations during fictive ecdysis accurately reflected the *in vivo* ecdysis motor behavior. To do so, we analyzed the pupal ecdysis motor behavior of intact animals removed from their puparium. The analytical process is divided into three steps: computation of the position of midline of the pupa, generation of the time-space diagram, and quantification.

To compute the midline of the pupa, a sequence of image processing operations was applied to every frame of the video (see Fig 9 and further details in Methods). The result is a time series indicating the position of the midline of the pupa with respect to the lateral axis, at each position along the antero-posterior axis.

The varying position of the midline was used to generate a time-space diagram, where time is mapped in the horizontal axis and the antero-posterior axis is mapped along the vertical axis. The color code indicates the position of the midline along the lateral axis. The diagram shows a distinctive pattern for each major motor pattern (Fig 10A). Peristaltic motor activity begins in the anterior and propagates to the posterior region of the animal, generating descending line patterns (from top-left to bottom right). The swinging motor pattern is characterized by a large variation of the midline position in the anteroposterior mid-section and a lengthening and shortening of the diagram across the anteroposterior axis. The lengthening occurs when the pupa is straight and the shortening when it bends to the side. Finally, the stretch-compression activity generates variation of the diagram across the anteroposterior axis as is seen during the swinging pattern, but with minimal variation of the midline position in the mid-section.

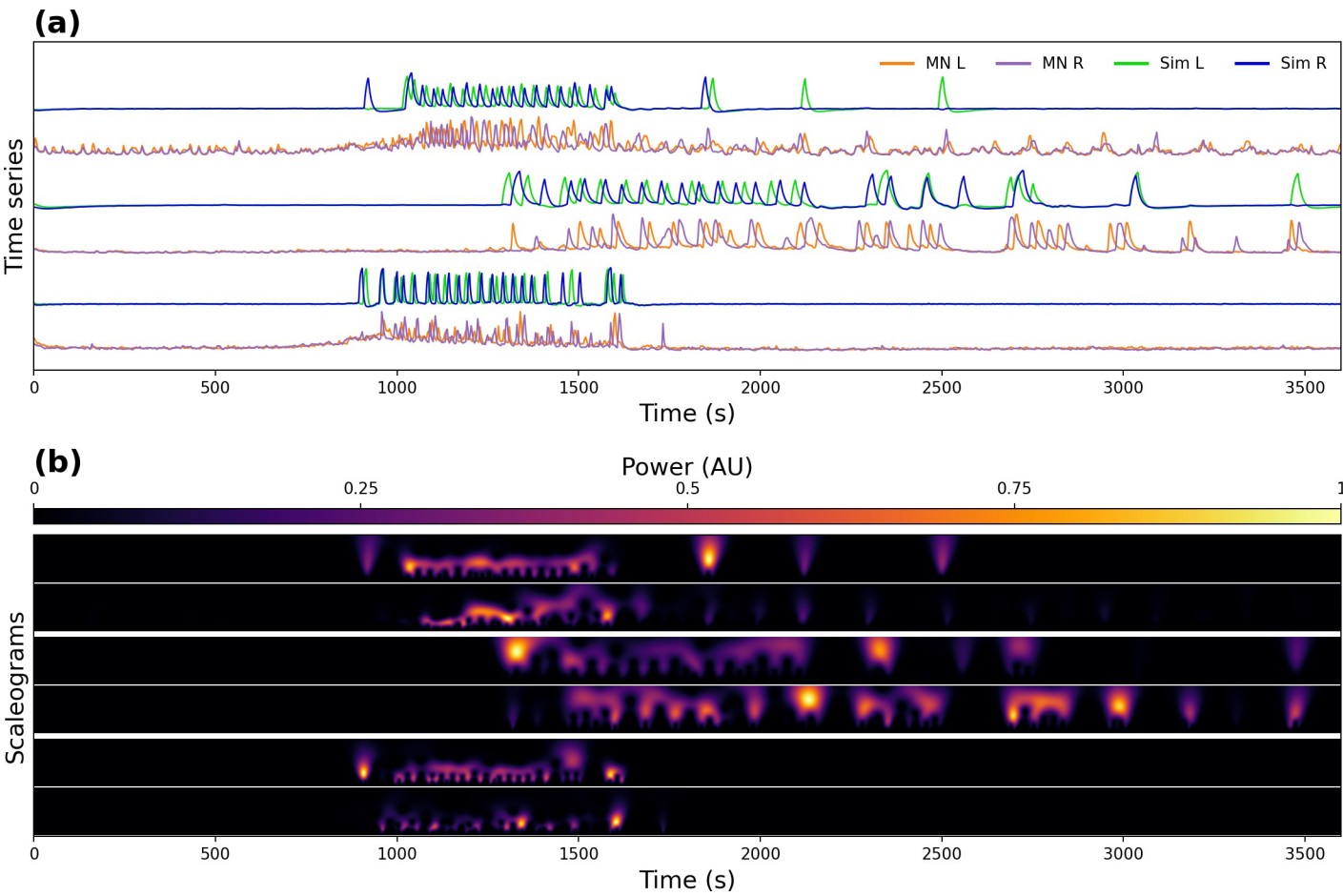

**Fig 8. Simulations of the Logistic model linked to a conductance-based bursting model. (a)** Experimental and simulated time series for 3 experiments. "MN L", "MN R", "Sim L" and "Sim R" indicate the left and right experimental, and left and right simulated motoneuron activity respectively. **(b)** Corresponding scaleograms, except that the scaleogram for the simulations is the average of 10 simulations. Each row shows the experimental (bottom) and the simulated (top) motoneuron activity, respectively.

We used this procedure to process 6 videos of pupal ecdysis behavior and generated the corresponding time-space diagrams (Fig 10B) and time series of the mid-sections (Fig 10C). Since pupae were not stimulated with exogenous ETH at the beginning of the video (as was the case for calcium recordings), we aligned the diagrams and time series to the beginning of the ecdysis phase.

All 6 time-space diagrams showed a similar pattern with very small differences in their timing and periodicity of oscillation. Quantifications were done manually by measuring time in the time-space diagram and time series plots (Fig 11). To measure the period of oscillation, we measured the duration of the largest time span of full cycles and divided it by the number of cycles. A swinging cycle was defined as a bending to one side followed by a bending to the opposite side. During post-ecdysis, a cycle included the bending to both sides followed by the stretch-compression motor pattern.

Finally, we compared the metrics obtained from the behavior of the intact pupal preparations to those of the motoneuron activity in the *ex vivo* CNS preparations (Fig 11).

The pre-ecdysis peristaltic contractions period of the pupa averaged 59.4 ± 8.7 s. We were able to visually detect the motoneuron peristaltic activity in the motoneuronal recordings, but

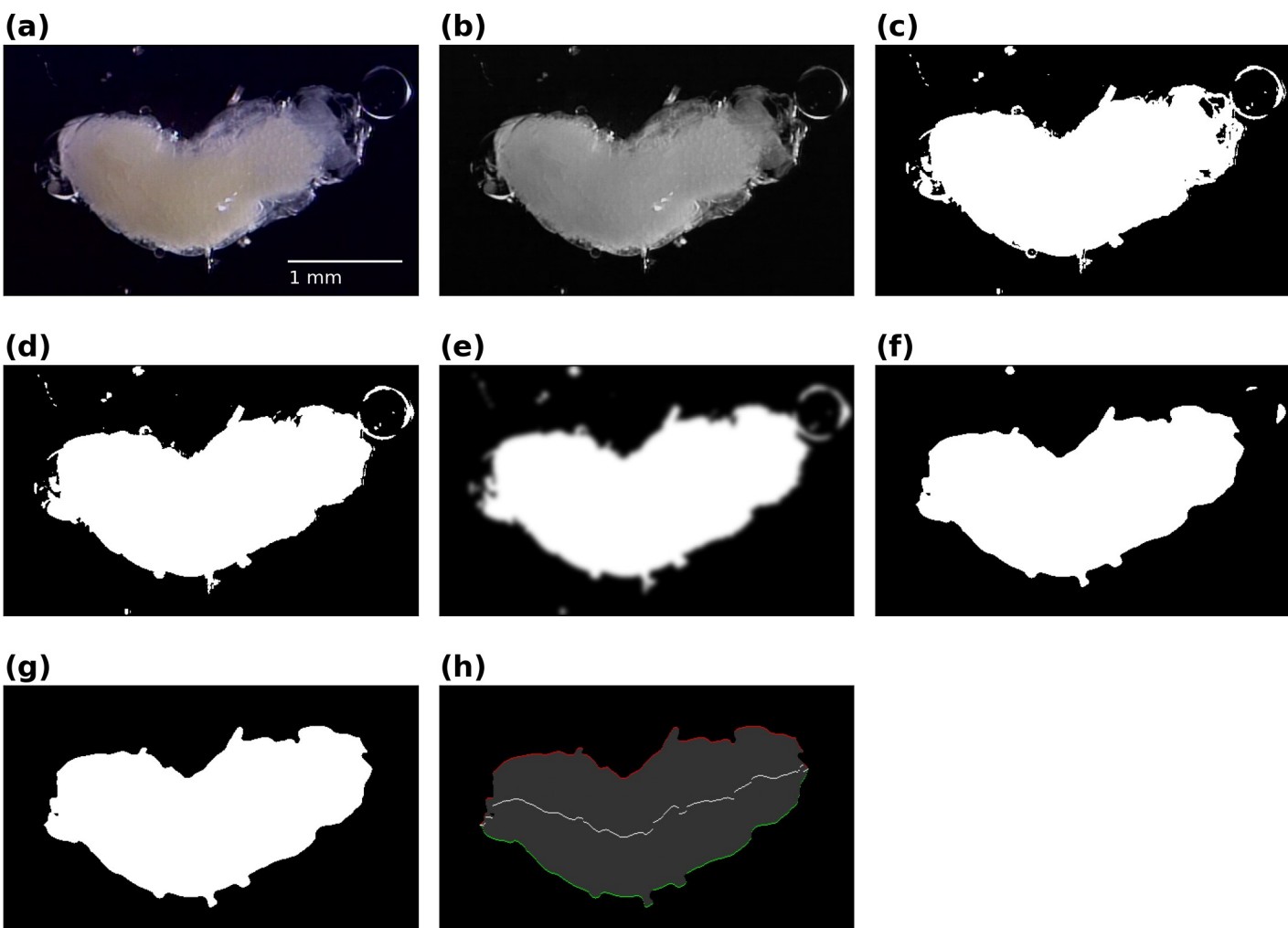

**Fig 9. Midline computation.** Sequence of image processing steps used to compute the midline of the pupa for every frame of the video. A RGB video frame is extracted **(a)** and converted to a grayscale image **(b)**, then thresholded **(c)**, and holes are then removed **(d)**. Blob borders are softened by applying a gaussian filter **(e)** and thresholded again **(f)**. Small blobs are discarded **(g)** and the left (green) and right (red) borders are computed **(h)**. The mean of the two borders represents the midline (white line).

because of the low SNR and time resolution, its quantification was not reliable. The ecdysis swinging contraction period was significantly shorter in the motoneuronal recordings than in the recordings of intact pupae ($25.1 \pm 2.4$ s versus $45.7 \pm 3.2$ s, respectively; p-value $< 0.002$, two-tailed Mann-Whitney U test). The mean duration of the ecdysis phase, on the other hand, was not significantly different ($346.7 \pm 37.8$ s versus $363.2 \pm 31.2$ s, respectively).

The mean post-ecdysis cycle period was $128.8 \pm 37$ s in the motoneuronal recordings. In intact animals, in contrast, we noticed that the post-ecdysis phase could be divided into two subphases: a fast one followed by a slow one. Both subphases included alternations between periods of swinging and periods of stretch-compression contractions, but in the slower phase the stretch-compressions tended to be of longer duration. The mean period was $76.4 \pm 2.4$ s for the fast and $187.3 \pm 9.6$ s for the slow subphases, respectively. We compared the mean of the three phases and found significant differences only between the slow and fast post-ecdysial phases (p-value $< 0.004$, two-tailed Mann-Whitney U test). The mean duration of the fast

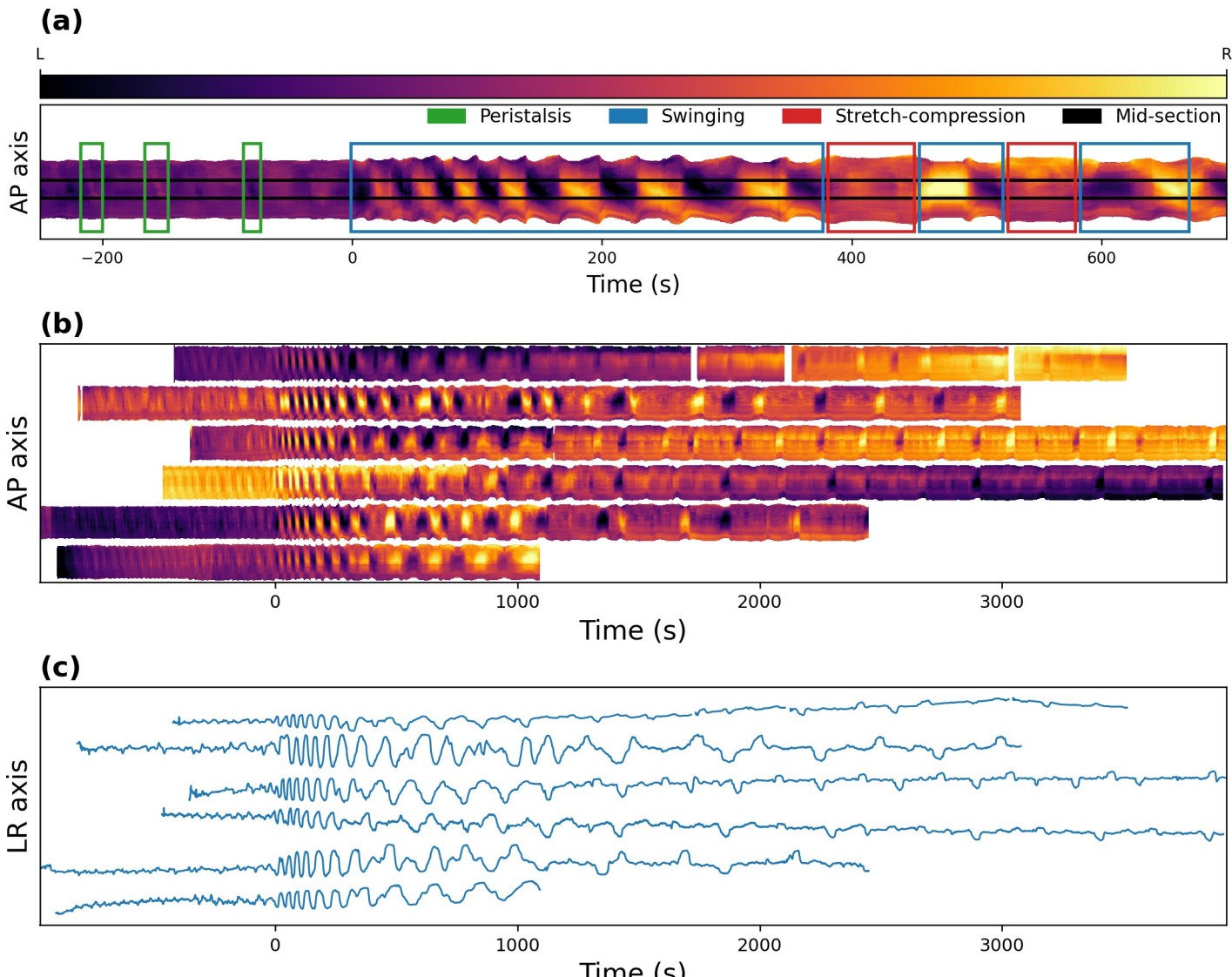

**Fig 10. Behavioral analysis. (a)** Time-space diagram pattern that allows the identification of three different motor routines. The "AP axis" represents the anteroposterior axis, which is oriented so that the top of the diagram corresponds to the anterior side of the pupa. The color indicates the position of the midline in the left-right axis along the anteroposterior axis, with the top and bottom of the diagrams corresponding to the anterior and posterior sections of the midline, respectively. Darker colors indicate that the midline section is closer to the left side, whereas lighter colors indicate that it is closer to the right side. **(b)** Filtered time-space diagrams for 6 pupal recordings aligned to the time when the ecdysis phase began. White spaces in the top diagram correspond to times when the pupa moved outside of the microscope viewing field. **(c)** Time series of the mid-section of **(b)** ("LR axis" represent the left-right axis).

post-ecdysis phase was 724.1 ± 22.1 s. The slow post-ecdysis phase duration could not be measured as it persisted past the end of the recording time.

In summary, we found that most of the activity recorded during pupal ecdysis behavior in intact animals had a fictive counterpart, but the *ex vivo* motoneuronal recordings showed different timing and greater variability. This indicates that the neural circuit controlling ecdysis behaves differently when tested in isolation, suggesting that sensory feedback could play an important role in regulating the timing of the ecdysis sequence.

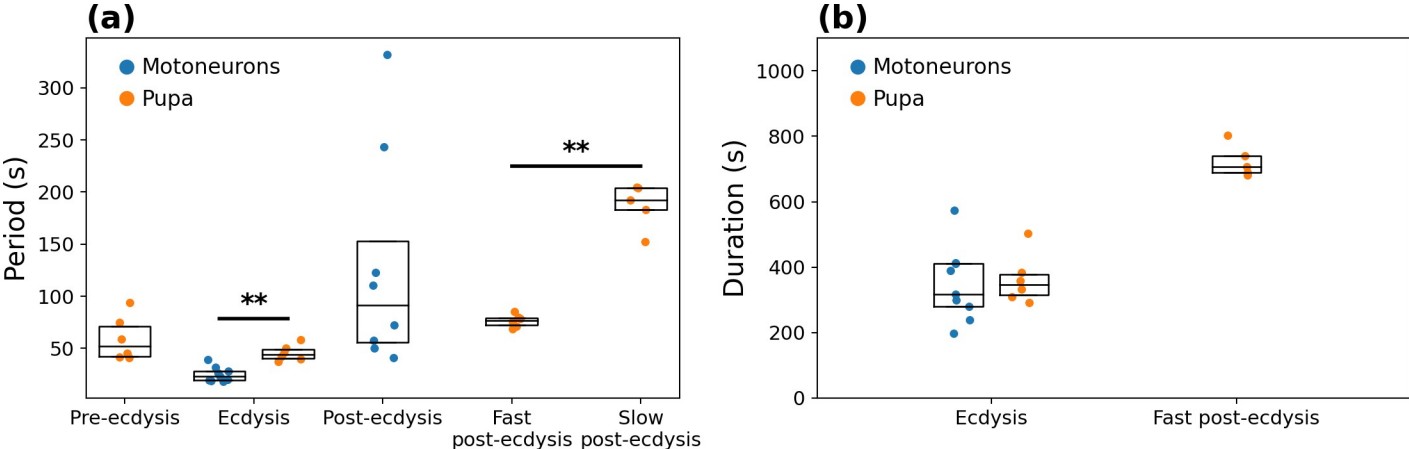

**Fig 11. Behavioral metrics.** Metrics and comparison of motoneuronal activity (n = 9) and pupal behavior (n = 6). **(a)** Period of the characteristic motor patterns of each of the ecdysial phases. **(b)** Duration of the ecdysis and of the fast post-ecdysis phase.

## Discussion

Ecdysis behavior consists of 3 separate motor programs (pre-ecdysis, ecdysis and post-ecdysis) that are expressed in a specific temporal order. A successful ecdysis is the result of the interplay between peptidergic neurons and motoneurons, each with quite different temporal patterns of activation. Here, we have combined calcium imaging recordings, computational tools, and behavioral analyses to gain better insights into the functional relationships between these two neuronal populations.

Our results suggest that CCAP neurons trigger motoneuronal activity and also sustain it throughout the ecdysis and post-ecdysis subroutines. Moreover, our analyses reveal a cross-frequency interaction between these two neuronal populations, as the slow variations in CCAP activity is correlated with the faster oscillations of motoneurons. This raises the question: What is the advantage of having CCAP neurons continuously modulating motoneuronal outputs? One possible answer is that it provides a continuous control over the desired behavior, allowing feedback mechanisms or environmental conditions to modify it. Our analyses show that the activity patterns of CCAP neurons correlate with different motor outputs: high CCAP activity generates swinging contractions over the ecdysis phase, whereas the alternation between high and low amplitude CCAP responses signals the transition from swinging contractions to stretch-compression movement, which are distinctive of post-ecdysis. Thus, specific CCAP activity patterns could be triggering the release of neuropeptides and potentially neurotransmitters in an activity-dependent manner to modulate diverse motor outputs. It has been reported that co-transmitters have distinct activity thresholds for their release, providing opportunities for circuit flexibility. For example, a low, tonic firing frequency may result in the release of neurotransmitters, whereas rhythmic bursting pattern may cause the release of both neurotransmitters and neuropeptides [37]. Consequently, a firing rate-dependent response would generate the modulation of diverse post-synaptic outputs [38–40]. To confirm that the CCAP activity pattern is modulating the release of neuropeptides or neurotransmitters in an activity-dependent way throughout ecdysis, an *in vivo* characterization of neuropeptide-neurotransmitter release associated to specific activity patterns and behavioral phases would have to be undertaken.

We showed that the activity of CCAP neurons can be correlated with motoneuron oscillatory activity using both an abstract logistic model and a conductance-based model. Using

these models, CCAP activity could accurately predict motoneuronal oscillatory activity (error rate <10%). In addition, the models showed that in general only 4 CCAP neurons or fewer (of a total of 8), were required to predict the start of an oscillatory episode. A possible explanation for this result is related to redundancy within the CCAP ensemble. In *Drosophila*, the CCAP AN1-AN4 network consists of 16 neurons that have been characterized from a molecular to a functional point of view [22,23,25,27]. The functional redundancy predicted by our model can have two interpretations. On one hand, it is possible that only one half of the CCAP AN1-AN4 cells may be necessary and sufficient to trigger ecdysis and post-ecdysis phases, with the rest of the network adding robustness and flexibility to the control of motoneuronal oscillations. Alternatively, all CCAP AN1-AN4 neurons may be necessary for the oscillatory command to reach the entire motoneuronal population. In this case, the statistical redundancy emerges just because of their coordination and synchronization. These two scenarios could be distinguished experimentally, through the selective inactivation of one or more CCAP neurons using recently developed holographic optogenetic tools [41–43].

With the addition of a simple conductance-based bursting CPG model [36,44], we were able to replicate the motoneuronal oscillatory activity. In the absence of any electrophysiological information about the neurons that generate this rhythm and the ion channels involved, we chose a simple generic neuronal oscillator. However, the well-known degeneracy and diversity found in oscillation-generating circuits [45–47] implies that many different neuronal models could produce the same output. It also implies that it is virtually impossible to make a better prediction without further electrophysiological evidence, and moreover, that the details of mechanisms found in different species can diverge significantly. Our analysis methods provide a framework to interpret new experimental manipulations that can be made to the mechanisms downstream the CCAP command signal.

Indeed, there are several neuropeptidergic modulators intervening in the generation of ecdysis behavior. We have previously shown that removing the action of key neuropeptides from the ecdysial neural system (such as EH, PBURS or ETH) causes significant abnormalities to the activity of the CCAP network and motoneurons, as well as dramatic defects to the behavior [25]. In the future, we can apply the analysis pipeline presented here to simultaneous CCAP and motoneuronal recording carried out in mutant flies lacking the expression of one of these peptides. In particular, it will be of interest to study the role of neuropeptide EH, because we have previously reported that the deletion of EH causes significant alterations to the temporal organization of the activity of the CCAP ensemble [25]. By using our quantitative approach, we would be able to better characterize the role of EH and of other neuropeptides in coordinating the temporal relationships between CCAP neurons in generating rhythmic motor outputs.

The 3 different motor programs of ecdysis can be observed in puparium-free preparations. Until now, these components have only been characterized qualitatively, on the bases of obvious motor changes [22,25]. Our computational method allowed us to quantitatively characterize the ecdysis and post-ecdysis behavioral programs and to contrast them with the associated motoneuronal activity. These analyses showed that most of the activity observed during pupal behavior had its fictive counterpart in the *ex-vivo* preparation. However, the patterns of motoneuronal activity were more variable compared to the behavior observed in the intact animal. This mismatch could be caused by sensory feedback, which has been shown to be critical for the proper organization of motor programs in many animals [33,48,49]. Sensory information may impact CCAP network activity itself or, alternatively, that of downstream CCAP targets. One possible source for this sensory (proprioceptive) feedback are somatosensory neurons located along the body wall of the pupa. In this regard, computational modeling of the neural circuits involved in the production of peristaltic waves during larvae crawling, have shown

that adding sensory feedback to a CPG network model affects both the speed and the intersegmental phase relationships [50]. Moreover, recent experimental work supports the idea that proprioceptive feedback plays a key role in the proper coordination of muscle contraction and in the speed of wave propagation [51]. Additional work must be done in order to identify the potential proprioceptive pathway that modulates the ecdysis motor sequence. Finally, biomechanical constraints imposed in the whole animal, which are absent in the puparium-free preparation, could further modify the resulting patterns of movement and produce greater spatiotemporal regularities.

Finally, our algorithm also detected that the post-ecdysis phase can be divided into fast and slow contraction frequencies. This newly detected motor program seems to be absent from our $Ca^{+2}$ imaging recordings, suggesting that additional neuronal layers may exist between CCAP and motoneurons, whose activity pattern has not yet been detected or characterized. Although this finding deviates from the main purpose of the present work, it is a good example of insights that can be obtained as a result of a better quantification of animal behaviors. Using our computational tools, future experimental research will be able to quantitatively relate this second post-ecdysis phase to the activity of other neural populations or gene expression networks.

## Methods

### Fly lines

*Drosophila melanogaster* cultures were raised on standard agar/cornmeal/yeast media and housed at 22–25˚C. The following GAL4 drivers were used: *Ccap*-GAL4 (driver for CCAP neurons; [52]) and C164-GAL4 (driver for motoneurons; [53]). We obtained the genetically encoded calcium sensitive, GCaMP3.2, from Julie Simpson (Janelia Farm, USA). GCaMP3.2 was expressed in CCAP neurons and motoneurons simultaneously by combining the *Ccap*-GAL4 and C164-GAL4 drivers using standard techniques.

### Imaging of calcium dynamics

Calcium ($Ca^{2+}$) recordings were carried out essentially as described in Mena *et al.* [25]. Briefly, animals containing a bubble in the mid-region of the puparium (~4 hours before pupal ecdysis) were selected. The central nervous system (CNS) was dissected in cold PBS, immobilized in 1.5% low melting temperature agarose solution (Sigma type VII-A; Sigma-Aldrich Chemical Co., MO) and covered with Schneider's Insect Medium (Sigma-Aldrich Chemical Co., MO). Recordings were performed using an Olympus DSU Spinning Disc microscope (Olympus Corporation, Shinjuku-ku, Tokyo, Japan) under a 20 X W NA 0.50 immersion lens. GFP signal was acquired using an ORCA IR2 Hamamatsu camera (Hamamatsu Photonics, Higashi-ku, Hamamatsu City, Japan) using the CellR Olympus Imaging Software (Olympus Soft Imaging Solutions, Munich, Germany). Fictive ecdysis was triggered by adding 600 nM of ETH1 (Bachem Co., USA). We recorded multiplane fluorescence using a sampling rate of 1 picture every 2–3 second for at least 60 min. Depending on the preparation, the number of images per z-stack was 3–5 focal planes (covering 100–200 μm in depth), which allowed the entire motoneuronal and CCAP network to be imaged.

### Video pre-processing

Video sequences were first processed using ImageJ [54]. Calcium time series were first detrended in order to compensate for slow variations in fluorescence during the recording caused by tissue drifting, then normalized in order to make the time series more uniform in

terms of the minima and maxima of fluorescence. Detrending was performed by finding the minimum values during the first and the last 250 s of the time series, generating a line that crossed those points, and subtracting the corresponding value from each frame. The normalization linearly mapped the time series so that the minimum and maximum values were 0 and 1, respectively.

$$m = \frac{f(t_1) - f(t_0)}{t_1 - t_0}$$

$$g(t) = f(t) - mt$$

$$h(t) = \frac{g(t) - g_{min}}{g_{max} - g_{min}}$$

Where $f(t)$, $g(t)$, $h(t)$ are the unprocessed, the detrended, and the preprocessed signals, respectively. $t_0$, $t_1$ are the time of the minimum value within the first and last 250 s of the signal, respectively, and $m$ the slope of the detrending line. $f_{min}$ and $f_{max}$ are the minimum and maximum value of the detrended signal, respectively.

We focused most of our analyses on the GCaMP activity of individual CCAP neurons of the $\alpha$ class [25], whereas for the motoneurons we analyzed the total GCaMP activity on the left and right sides of the abdominal CNS, due to their large number. In preparations in which both CCAP and motoneurons expressed GCaMP, the 2 sets of neurons were readily distinguishable by their position and size.

## Activity onset

The computation of activity onset was performed using a smoothened version of the time series, obtained by convolving the original time series with a rectangular window function of 10 s duration and area of 1. Onset was defined as the first instant for which the convolved time series exceeded 1/2 of the maximum amplitude of the original time series. The first 100 s of the time series were discarded because some neurons displayed a high level of fluorescence at the beginning of the recordings.

## Time frequency analyses

Time-frequency analyses were performed using the continuous wavelet transform (CWT) [55] with the complex Morlet wavelet ($\sigma = 3$). The CWT is defined by:

$$W(t, s) = \frac{1}{s} \int_{-\infty}^{\infty} f(u) \overline{\psi} \left( \frac{u - t}{s} \right) du$$

Where $s$ is the scale parameter, $t$ the position parameter, $f()$ the signal function, $\psi()$ the wavelet function and the overline represents the complex conjugate.

The complex Morlet is defined as:

$$\Psi_\sigma(t) = \left( 1 + \exp(-\sigma^2) - 2\exp\left( -\frac{3}{4}\sigma^2 \right) \right)^{-\frac{1}{2}} \pi^{-\frac{1}{4}} \exp\left( -\frac{1}{2}t^2 \right) \left( \exp(i\sigma t) - \exp\left( -\frac{1}{2}\sigma^2 \right) \right)$$

Which has a central frequency $\sim\sigma$ or central period $\sim 1/\sigma$.

The scaleogram (analog to the spectrogram) is defined as the square of the amplitude of the CWT:

$$X(t, s) = W(t, s)\overline{W(t, s)}$$

The scale is related to the period ($T$) in the following relationship:

$$T =\sim s/\sigma$$

## Conductance based model

We adapted a CPG model developed by Jalil et al. (2010) by adding the $I_{i,CCAP}$ and $I_{i,X}$ terms, which represent currents generated by input from CCAP neurons and stochasticity, respectively; and by adding an equation that models the calcium fluorescence induced by neuronal activity.

$$\frac{dV_i}{dt}(t) = -\frac{I_{i,\text{Na}} + I_{i,\text{K}} + I_{i,\text{L}} + I_{i,\text{Syn}} + I_{i,\text{CCAP}}(t) + I_{i,\text{X}}}{C} \tag{1}$$

$$I_{i,\text{Na}} = g_{\text{Na}}(V_i - E_{\text{Na}})n_i^3 h_i \tag{2}$$

$$I_{i,\text{K}} = g_{\text{K}}(V_i - E_{\text{K}})m_i^2 \tag{3}$$

$$I_{i,\text{L}} = g_{\text{L}}(V_i - E_{\text{L}}) \tag{4}$$

$$I_{i,\text{Syn}} = g_{\text{Syn}}(V_i - E_{\text{Syn}})s(-1000(V_j + 0.0225)), \quad i \neq j \tag{5}$$

$$I_{i,\text{CCAP}}(t) = g_{\text{CCAP}}(V_i - E_{\text{CCAP}})p(t) \tag{6}$$

$$dI_{i,\text{X}} = -\frac{I_{i,\text{X}}}{\tau_{\text{X}}} + \sigma_{\text{X}}W_{i,t} \tag{7}$$

$$n_i = s(-150(V_i + 0.0305)) \tag{8}$$

$$\frac{dh_i}{dt} = \frac{s(500(V_i + 0.0333)) - h_i}{\tau_{\text{Na}}} \tag{9}$$

$$\frac{dm_i}{dt} = \frac{s(-83(V_i + V_{\text{Shift}})) - m_i}{\tau_{\text{K}}} \tag{10}$$

$$\frac{df_i}{dt} = \frac{s(-100(V_i + 0.04)) - f_i}{\tau_{\text{f}}} \tag{11}$$

$$s(x) = \frac{1}{1 + \exp(x)} \tag{12}$$

In this model, $V$ is the membrane voltage, $C$, the membrane capacitance, and $t$, the time. $I$, $g$, $E$, $\tau$, represent, respectively, the current, the maximum conductance, the reversal potential, and the time constant. The subscripts $i$, $j$ refer to neuron index, Na, K, L, Syn, CCAP, X refer to sodium, potassium, leakage, synapse, CCAP, and noise (which is an Ornstein-Uhlenbeck

process), respectively. $p(t)$ is the motoneuronal oscillation probability. $W_t$ represents a Wiener process and $\sigma_x$ its volatility. $n$, $h$, $m$ are the sodium activating, sodium inactivating, and potassium activating gating variables, respectively; $f$ represents the calcium imaging fluorescent intensity. $V_{\text{Shift}}$ is a potassium activation curve shifting parameter.

Eq 6 was added to generate a depolarizing current during the predicted oscillatory phase. $p(t)$, the oscillating probability at time $t$ from the logistic model, approaches 0 when the oscillation probability is low and 1 when it is high. In the conductance model, $p(t)$ acts as a gating variable, $g_{\text{CCAP}}$ is the maximum conductance parameter and $E_{\text{CCAP}}$ is the reversal potential. A high $p(t)$ value generates depolarizing currents that help the system reach the voltage threshold to fire action potentials (APs) in a bursting and alternating pattern between the neurons of the circuit. By contrast, a low $p(t)$ value tends to keep the system in a non-oscillatory state.

Eq 7 adds stochasticity to the model, which has the effect of mimicking the probabilistic influence of the α CCAP neurons on the motoneuronal oscillatory activity. It also adds phase noise during the oscillatory activity, similar to the one observed in the experimental recordings (Fig 4).

Eq 11 generates fluorescence (calcium) spikes, which respond with a time constant $\tau_f$. The equation produces increases in the values of fluorescence during the bursting phase and decreases during the non-bursting phase.

Simulations were done using the following parameters: $C = 0.5$ nF; $\tau_{\text{Na}} = 0.055$ s; $g_{\text{Na}} = 200$ nS; $g_K = 45$ nS; $g_L = 10$ nS; $g_{\text{Syn}} = 0.5$ nS; $g_{\text{CCAP}} = 1$ nS; $E_{\text{Na}} = 0.045$ V; $E_K = -0.07$ V; $E_L = -0.046$ V; $E_{\text{Syn}} = -0.0625$ V; $E_{\text{CCAP}} = 0$ V; $V_{\text{Shift}} = 0.022$ V; $\tau_X = 0.001$ s; $\sigma_X = 0.03$ nA.

The values for $\tau_K$ and $\tau_f$ varied as they were adjusted to each individual experiment. Their values are given in Table 2

The oscillation period of the calcium spikes is affected by many parameters, but $\tau_K$ affects it linearly and does not affect the duty cycle and phase difference, making it much easier to adjust manually. The exponential decay on the other hand can only be fitted through $\tau_f$ and does not affect the model dynamics.

## Time constants

The time constant for the fluorescence signal was computed by fitting an exponential decay function to the data, defined by:

$$f(t) = a \exp\left(\frac{-t}{\tau}\right) + b$$

Where $a$, $b$ and $\tau$ are constants, $a$ represents the amplitude of the decay, $b$ the basal fluorescence, and $\tau$ the time constant. The time constant of a segment was defined to be the $\tau$ of the fitted 15 s template, which was adjusted through the least squares method.

The average time constant ($\tau_f$) of the exponential decay of motoneuron calcium spikes was computed using only high quality (high SNR) segments of the recordings.

**Table 2. Values of $\tau_K$ and $\tau_f$ used for the fit of each experiments.**

| Parameter | Experiment | | | | | | | | |
|---|---|---|---|---|---|---|---|---|---|
| | 1 | 2 | 3 | 4 | 5 | 6 | 7 | 8 | 9 |
| $\tau_K$ (s) | 55.2 | 63.9 | 125.6 | 67.7 | 136.5 | 71.5 | 71.5 | 69.0 | 88.1 |
| $\tau_f$ (s) | 3.2 | 4.1 | 11.2 | 10.0 | 11.2 | 4.8 | 8.6 | 3.7 | 4.4 |

The time constant of the potassium gating variable ($\tau_K$) was adjusted manually to make the periodicity of oscillations of the model match the previously measured periodicity of motoneuronal oscillation.

## Puparium-free behavioral recordings and processing

Behavioral recordings were carried out as described in Mena et al., 2016 [25]. Briefly, the pupa was surgically removed from the puparium at the very start of pre-ecdysis, placed in a recording chamber, and covered with halocarbon oil (Sigma-Aldrich Chemical Co., MO). The animals were filmed under transmitted light using a Leica DMLB microscope (20 X magnification) for at least 60 min.

Color RGB frames were extracted from the video sequence in real number format, with 0 and 1 representing the minimum and maximum intensity, respectively (Fig 10A). The images were converted to grayscale by averaging the intensity of the three-color component channels (Fig 10B). Pixels of the images were thresholded by setting them to 0 if the grayscale values were lower than 0.1 or to 1.0 otherwise (Fig 10C). Holes, or black regions inside white regions were then removed (Fig 10D). Although the value of the threshold was chosen arbitrarily, we found that the resulting thresholded image was not very sensitive to its exact value. To extract the main behavioral features, high spatial frequency details were removed in a two-step process. The images were first convolved with a gaussian function with σ = 3 pixels (Fig 10E) and then thresholded at a threshold of 0.5, to avoid expanding (dilating) or reducing (eroding) the borders of the pupa (Fig 10F). Regions with less than 10000 white pixels, roughly 20% of the area of the pupa (see Fig 10G), were discarded. The left- and rightmost pixels of the pupa along the anteroposterior axis were then computed. The average between the left- and rightmost pixels along the axis were considered to represent the midline of the pupa (Fig 10H).

## Supporting information

**S1 Fig. Motoneuronal coordination.** (a) Projection of 5 images from different planes, of GCaMP3.2-expressing CCAP neurons and motoneurons. (b) Time series for each motoneuron and average for each side. (c) Correlation matrix of the motoneuron activity traces, showing the highest correlation between motoneurons of the same side.
(PDF)

**S1 Video. Example of behavioral analysis.** The video shows the original raw image of an intact pupa during the ecdysis behavior (left) and the results of the image processing described in Fig 9 (right). The bottom panel shows the position of the midline in the left-right axis, as described in Fig 10.
(MP4)

## Author Contributions

**Conceptualization:** Miguel Piñeiro, Wilson Mena, John Ewer, Patricio Orio.

**Data curation:** Miguel Piñeiro, Wilson Mena.

**Formal analysis:** Miguel Piñeiro.

**Funding acquisition:** John Ewer, Patricio Orio.

**Investigation:** Miguel Piñeiro, Wilson Mena.

**Methodology:** Miguel Piñeiro, Wilson Mena.

**Project administration:** Patricio Orio.

**Resources:** John Ewer, Patricio Orio.

**Software:** Miguel Piñeiro.

**Supervision:** John Ewer, Patricio Orio.

**Validation:** Miguel Piñeiro, Wilson Mena, John Ewer, Patricio Orio.

**Visualization:** Miguel Piñeiro.

**Writing – original draft:** Miguel Piñeiro, Wilson Mena.

**Writing – review & editing:** Miguel Piñeiro, Wilson Mena, John Ewer, Patricio Orio.

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
