## [Decision Letter · Decision Letter 0]

19 Jun 2021

Dear Mr. Orio:

Thank you very much for submitting your manuscript "Extracting temporal relationships between weakly coupled peptidergic and motoneuronal signaling: application to Drosophila ecdysis behavior" (PCOMPBIOL-D-21-00575) for review by PLOS Computational Biology. 

As with all papers reviewed by the journal, your manuscript was reviewed by members of the editorial board and by two independent reviewers. Based on the reviews, we regret that we will not be pursuing this manuscript for publication at PLOS Computational Biology.

Indeed, the reviewers did appreciate many aspects of the work. However, in addition to their concerns listed below, the modeling is mainly marginal to the study and thus is not really appropriate for PLoS Comp Biology. 

The reviews are attached below this email, and we hope you will find them helpful if you decide to revise the manuscript for submission elsewhere. 

While we cannot consider your manuscript further for publication in PLOS Computational Biology, we would like to offer you the option to transfer your submission, with reviews, to PLOS ONE https://www.editorialmanager.com/PONE/

If you DO wish to transfer your submission, please click this link:

<DeepLinkData><DeepLinkTypeID>27</DeepLinkTypeID><peopleID>393233</peopleID><userSecurityID>b5928cd4-7955-422d-9263-82f412c41bed</userSecurityID><documentID>29386</documentID><revision>0</revision><manuscriptNumber>PCOMPBIOL-D-21-00575</manuscriptNumber><docSecurityID>3c4b334f-de06-4c95-81c5-7fd99113c024</docSecurityID></DeepLinkData>

If you do NOT wish to transfer your submission, please click this link to decline:

<DeepLinkData><DeepLinkTypeID>28</DeepLinkTypeID><peopleID>393233</peopleID><userSecurityID>b5928cd4-7955-422d-9263-82f412c41bed</userSecurityID><documentID>29386</documentID><revision>0</revision><manuscriptNumber>PCOMPBIOL-D-21-00575</manuscriptNumber><docSecurityID>3c4b334f-de06-4c95-81c5-7fd99113c024</docSecurityID></DeepLinkData>

Please note, all PLOS journals are editorially independent and vary in submission requirements.

Should you choose to transfer, your manuscript files, along with the reviewers' comments and their identities will be transferred automatically, and you will receive a confirmation email within 24 hours. Once transferred, your submission will be returned to you so you can check over your record before completing the submission. You may be asked to provide additional information, such as a response to the reviewers' comments. If you have any questions, please contact the editorial office of PLOS ONE https://www.editorialmanager.com/PONE/

We are sorry that the news is not more positive on this occasion, and we hope you will consider PLOS Computational Biology for future submissions. Thank you for your support of PLOS and of open-access publishing.

Sincerely,

Lyle Graham

Deputy Editor

PLOS Computational Biology

Reviewer's Responses to Questions

**Comments to the Authors: **

Reviewer #1: This manuscript considers how the temporal aspects of neuromodulation are involved in the production of behavior. Specifically, the authors seek to address the role of temporal modulation in the control of Drosophila ecdysis by describing the influence of peptidergic modulation from CCAP-releasing neurons on motor neurons. Towards this, the authors simultaneously measure the population activities of CCAP and motor neurons during fictive ecdysis behavior in ex-vivo preparations using calcium imaging, and report that the peptidergic neuron population activity is tightly coupled to the activity of the faster motor neuron population.Computational models are constructed to capture the calcium dynamics of CCAP neurons, and simulations demonstrate that these models mostly recapitulate experimental observations. Further, the authors develop an algorithm to classify the stages of in-vivo ecdysis behavior. These stages are then compared to ex-vivo motor neuron activity. The significance of the work is the attempt to link the influence of peptidergic modulatory neuron activity on motor neurons with behavior.

Major Comments

The manuscript is missing key information about methodology that severely limits the interpretation of the results.

The calculation of activity onset doesn’t appear to be defined in the manuscript.

Figure 1 c suggests that the signal of L and R motor neurons are being grouped. 

The claim that there are no major differences in the activity between sides is not supported in the text. Please include a reference or a panel that supports this point.

Why are L and R motor neurons being grouped? The analysis should work with the individual motoneuron data points. 

Also, the one-tailed Mann-Whitney U test is sensitive to the variance of the data which would be under-represented when averaging the L and R motor neurons. 

The modeling component of this work is interesting but does not contribute to any point made in the manuscript. The authors should better explain the rationale for the models.

Mainly, addressing the utility of the ability to predict motor neuron activity from modeled CCAP activity using ad hoc tuned models. 

Maybe these models are useful to predict behavioral state transitions? If true, this should be highlighted and so far isn’t clear in the text.

ln 414: Why is there an attempt to classify motor neuron calcium recording using the results of the behavior algorithm? 

Behavior is an output that integrates multiple signals, and is usually a nonlinear function of motor neuron output.

Variability is expected to decrease from motor neuron activity to behavior output simply because there are fewer degrees of freedom in the biomechanics.

The finding about post-ecdysis phase being divided into fast and slow contraction frequencies as currently presented in the results and discussion (ln 495-500) does not contribute to the apparent main point of the paper (determining the interactions between peptidergic modulation and production of behavior)

Suggest to either better integrate for clarity or remove to prevent diluting the focus of the manuscript.

Minor Comments

 Line 65-66: “Ecdysis begins with the release of Ecdysis Triggering Hormone (ETH) into the circulatory system (hemolymph)...”

Line 78: The methods listed (PCA, ICA, SVD, k-means) are thought to be fairly general. What is meant by “restricted to specific datasets”?

Line 79: Offering more specific reasons for the introduction of new methods would strengthen the argument. See previous point.

Line 80: It’s not clear what is new about the computational approaches -- many of these approaches have been around for some time. 

Line 87: “... simulates many of the observed experimental features.”

Line 93: Replace “through” with “throughout”

Line 116 - 118: Unclear what is being grouped (117), when motoneurons are being divided into left and right regions.

Line 120: Perhaps this is a term that I’m unfamiliar with, but unsure what is meant by “ecdysial phase proper”

Line 123: What do the +/- indicate? Variance, SD, SE?

Lin 128: While technically true, not a completely fair statement since figure 1c shows 4 out of 9 experiments demonstrating the opposite

 Line 150: “In contrast” instead of “By contrast”

Line 368: Repeated “Fig 9” 

There is little discussion about how the activity of other modulatory neurons may be involved. This is not a deal-breaker, but what is known should be presented to better frame the results (ex. What other modulatory projections exist, how might that influence the interpretation of results)

Figure 1: 

1a and 1b are missing scale bars

1d:

It is not clear what the vertical bars indicate. May be more informative to represent this data as a box and whiskers plot

Please indicate precisely what is meant by “onset” -- a simple schematic or indication on 1c would help.

Reviewer #2: I consider that this paper contains significant novelty in the context of the study of the relationship between slow dynamics of neuromodularoty neurons and motoneurons and delivers both experimental and mixed experimental and modeling results that contribute to understand this relationship. Thus, in my opinion the paper is suitable for publication in PLoS Computational Biology. However, I find several issues that could be addressed to improve the paper:

1-It is clear from the timescales shown in the figures that the correlation analysis focuses on slow dynamics. Authors could comment on the slow time scale nature of the calcium imaging of both motoneurons and CCAP neurons. Is this the only relevant time scale of the interaction? Please discuss if motoneuron voltage activity is known to faster than calcium dynamics.

2-Please specify the number of recordings for each experimental trial in the statistics shown in Fig. 1d. 

3-Please explain how did you group the neurons to increase the signal to noise ratio.

4-The paper is focused on the frequency analysis, but regarding the coordination of motor movements it would be more useful an analysis of the sequences that built up the motor program (beyond the phase analysis), i.e., what is the effect of the neuromodulation in the sequential activation of the motoneurons that shape the behavioral observations. This could be at least discussed at the end of the manuscript, as a limitation of the proposed analysis and modeling approach if it cannot be addressed.

5-It is not very surprising that the logistic model captures the correlation from the model design.

6-How could the proposed methodology be used in other experimental models. This could be important to target a general readership.

Minor issues:

6-Why is real-time calcium-imaging mentioned in the abstract. Why is the real-time nature of the calcium imaging important. I think this can be misleading for some readers.

7-There are several problems with the dynamic references to the figures, e.g. lines 237, 254, 274, 291, 317, 335, 404, 415.

8-Please provide an example video of the behavioral analysis.

**Have the authors made all data and (if applicable) computational code underlying the findings in their manuscript fully available?**

Reviewer #1: Yes

Reviewer #2: Yes

PLOS authors have the option to publish the peer review history of their article (what does this mean?). If published, this will include your full peer review and any attached files.

Reviewer #1: No

Reviewer #2: No

---

## [Decision Letter · Decision Letter 1]

14 Nov 2021

Dear Mr. Orio,

We are pleased to inform you that your manuscript 'Extracting temporal relationships between weakly coupled peptidergic and motoneuronal signaling: application to Drosophila ecdysis behavior' has been provisionally accepted for publication in PLOS Computational Biology.

Best regards,

Lyle Graham

Deputy Editor

PLOS Computational Biology

Reviewer's Responses to Questions

**Comments to the Authors:**

Reviewer #2: The new version of the paper has addressed my main concerns. The paper is ready for publication. Thank you for your work.

**Have the authors made all data and (if applicable) computational code underlying the findings in their manuscript fully available?**

Reviewer #2: Yes

PLOS authors have the option to publish the peer review history of their article (what does this mean?). If published, this will include your full peer review and any attached files.

Reviewer #2: No

---

## [Editor Report · Acceptance letter]

25 Nov 2021

PCOMPBIOL-D-21-00575R1 

Extracting temporal relationships between weakly coupled peptidergic and motoneuronal signaling: application to Drosophila ecdysis behavior

Dear Dr Orio,

I am pleased to inform you that your manuscript has been formally accepted for publication in PLOS Computational Biology. Your manuscript is now with our production department and you will be notified of the publication date in due course.

With kind regards,

Livia Horvath
